# Contrast Variation Small Angle Neutron Scattering Investigation of Micro- and Nano-Sized TATB

**DOI:** 10.3390/ma12162606

**Published:** 2019-08-16

**Authors:** Panqi Song, Xiaoqing Tu, Liangfei Bai, Guangai Sun, Qiang Tian, Jian Gong, Guiyu Zeng, Liang Chen, Lili Qiu

**Affiliations:** 1School of Chemistry and Chemical Engineering, Beijing Institute of Technology (BIT), Beijing 100081, China; 2Key Laboratory of Neutron Physics and Institute of Nuclear Physics and Chemistry, China Academy of Engineering Physics, Mianyang 621999, China; 3Institute of Chemical Materials, China Academy of Engineering Physics, Mianyang 621900, China

**Keywords:** SANS, contrast variation method, TATB, micro/nano-energetic materials

## Abstract

Small angle neutron scattering (SANS) with contrast variation was used to characterize the fractal behavior and embedded porosity of micro/nano-sized 1,3,5-triamino-2,4,6-trinitrobenzene (TATB) crystallites, gauging the effects of particle sizes on the microstructural features. Scattering results reveal that the external surface of micro-sized TATB crystallites are continuous and smooth interfaces and their internal pores display a surface fractal structure (surface fractal dimension 2.15 < D_S_ < 2.25), while the external surface of nano-sized TATB particles exhibit a surface fractal structure (surface fractal dimension 2.36 < D_S_ < 2.55) and their internal pores show a two-level volume fractal structure (large voids consist of small voids). The voids volume fraction of nano-sized TATB particles are found increased distinctively when compared with micro-sized TATB particles on length scale between 1 nm and 100 nm. Specific surface areas are also estimated based on Porod law method, which are coincident with Brunauer-Emmett-Teller (BET) measurements. The contrast variation technique distinguishes the information of internal voids from external surface, suggesting SANS is a powerful tool for determining the microstructural features, which can be used to establish the relationship between microstructures and properties of micro/nano-energetic materials.

## 1. Introduction

Micro/nano-energetic materials, as a new type of functional materials, have attracted extensive attention due to their excellent performances, such as high energy releasing rate, exceptional combustion efficiency, tailored burning rate, increased shock sensitivity and reduced impact sensitivity [1,2,3]. As widely reported, microstructures such as particle morphologies, sizes, and void size distributions have significant influence on these properties [3,4,5,6,7,8]. It has been reported that embedded submicron pores play an essential part in the formation of hot spots, and groups of small pores can just produce the same efficiency compared with a single large pore in generating a hot spot [9,10,11]. Additionally, the fractal characteristics are an important factor influencing the sensitivity of different explosives [12]. Likewise, the total surface area is closely related to the presence of pores or cracks and affects the rate of combustion and heat transfer during combustion process. Therefore, precise characterization of the surface and void features via various types of apparatuses is essential for exploring the structure–property relationship of micro/nano-energetic materials.

For micro/nano-energetic crystallites grown in solutions, there are several types of microstructures, e.g., fractal/non-fractal (or smooth/rough) interfaces, external open pores, and embedded closed submicron pores caused by solvent inclusions, etc. [13]. Generally, a scanning electron microscope (SEM) is used to determine the surface features [14], while gaseous adsorption (BET) and mercury intrusion [15,16,17] methods are used to measure the porosity and specific surface area. It should be noted that SEM can provide qualitative information on the external morphologies (particle size, shapes and smooth or rough surface). BET and mercury intrusion can provide information on open pores. However, none of the above three methods are good for determining the fractal behaviors of the internal pores. In comparison, the small angle neutron scattering (SANS) method is a proven method to obtain the quantitative information of the fractal features and the pore structures [18,19,20,21,22], including both the open and closed voids. Furthermore, the method of contrast variation can separate the information involved with external structures from that involved with internal structures of complex systems [23,24,25]. In addition, SANS method is nondestructive, simple, and no special treatment of the sample is required, and does not put explosives in jeopardy of high temperature and high pressure. Therefore, SANS is a promising means to investigate the microstructures of micro/nano-energetic materials. 

In this paper, in order to study the effect of particle sizes on the microstructural features of micro/nano-sized 1,3,5-triamino-2,4,6-trinitrobenzene (TATB) crystallites, SANS measurement with contrast variation was conducted on four batches of TATB samples with two different particle sizes. Combined with the testing results of SEM and BET, we obtained the detailed fractal behaviors of the external surface and internal voids, the fractal structure size on length scales between 1 nm and 100 nm, and the specific surface areas. These data indicate that the particle sizes greatly influence the microstructural features of micro/nano-sized TATB and the obtained quantitative physical parameters can be used to establish the relationship of structure–property.

## 2. Theoretical Background of SANS and Contrast Variation Method

Small angle neutron scattering is due to the spatial modulation of neuron scattering length density (SLD) on the nanometer scale [25], which can reflect microscale fluctuation of the chemical and isotopic composition. Based on the SANS measurement, the contrast variation method can be used to separate the scattering signal of the shape (defined as external surfaces and surface defects) from that of the internal structure [16,20], which is achieved by adjusting the deuteration level of the solvent. Fluid mixtures with different amounts of deuterium are formulated to match the neutron SLD level of the TATB. Thus, Δ*ρ*, the contrast of the average SLD between the measured particles and the surrounding media, can be varied continuously to enhance or suppress different structural information. In this case, at one extreme, ∆*ρ*→0 (the contrast match point), the scattering signal mainly comes from the internal structure. At the other extreme, ∆*ρ* ≈ infinite (far from the contrast match point), the shape of the fluid-excluding parts of an object dominate the overall scattering. At any other level of contrast, both the shape and internal structure make an essential of the observed scattering [16]. Therefore, the scattered intensity can be described as a function of ∆*ρ* by the following expression [16]: (1)$$I(\Delta \rho ,Q)=\Delta \rho^{2}I_{\Omega }(Q)+\Delta \rho I_{\Omega \varsigma }(Q)+I_{\varsigma }(Q),$$
where $I_{\Omega }(Q)$ represents the scattered intensity from the shape of the solvent exclusion regions, $I_{\Omega \varsigma }(Q)$ describes the scattered intensity due to correlations between the shape and internal structure, and $I_{\varsigma }(Q)$ represents the scattered intensity from the internal structure of the solvent exclusion regions. 

Equation (1) is used to fit the experimental SANS data to obtain the basic scattering functions. It should be mentioned that Equation (1) is suitable for a chemically homogeneous system. This rule is valid in TATB, which contains a negligible amount of impurities.

## 3. Materials and Methods 

### 3.1. Materials

The TATB loose powders used in the current study are provided by the Institute of Chemical Materials, CAEP, China. In order to investigate how the particle size affects the microstructural features, two batches of micro-TATB loose powders (μTATB-1, μTATB-2, purity 99%, particle size about 14 μm) and two batches of nano-sized TATB loose powders (nTATB-1, nTATB-2) were used in the current study. 

### 3.2. Method

#### 3.2.1. Swelling Method

Approximately 100 mg of each specimen were loaded into a 1 mm path-length quartz cell with the methanol/deuterated methanol solvent and the cell was gently shaken to ensure even filling. The volume fill ratio (0.32–0.36) is calculated from the fill density, which is used for scattering intensity correction. TATB is insoluble in methanol and most of the other conventional solvents, so the change in neutron SLD of TATB caused by the hydroquinone exchange reaction at the solid–liquid interface is negligible. All the SLD can be calculated based on chemical formula and theoretical mass density of the components. Deuterated/nondeuterated methanol was chosen as the contrast solution because large contrast range is achievable (SLD = 5.80 × 10^10^/−3.73 × 10^9^ cm^−2^), and the surface tension of the fluid is smaller. Figure 1 depicts the SLD of the mixture of methanol and deuterated methanol as a function of volume fraction of deuterated methanol, *φ* D-methanol. The conditions of the experiments are marked by solid squares and the ratio of H-methanol/D-methanol (CH_3_OH/CD_3_OD) solvent are 0.00/1.00, 0.10/0.90, 0.15/0.85, 0.60/0.40, 1.00/0.00, respectively. The SLD of the mixtures varied from approx. −3.73 × 10^9^ to 5.80 × 10^10^ cm^−2^ by changing the volume fraction of the deuterated methanol. Note that the SLD level of the TATB falls within the range of the mixtures in the current experiment. Such measurements can make sure that the acquirement of scattering functions is more reliable.

#### 3.2.2. SANS Method

SANS experiments were performed on the Suanni small-angle neutron spectrometer at CMRR (China Mianyang Research Reactor) [26]. The scattered intensity *I*(*Q*) is measured as a function of scattering vector Q=(4π/λ)sinθ, where θ is half of the scattering angle and λ is the neutron wavelength. A multiblade mechanical velocity selector was used to obtain a monochromatic beam with a mean wavelength of 0.53 nm with a spread (Δλ/λ) of ca. 10%. The scattering data was collected in transmission mode by a ^3^He gas-filled multiwire detector with the sensitive area of 64 × 64 cm^2^. The sample-to-detector distances were 1.15 m, 4.29 m and 10.44 m to span the range of scattering vectors Q from 0.065 to 3 nm^−1^. The isotropic raw data were reduced to one dimension by BerSANS software (Hahn-Meitner-Institut, Berlin, Germany) [27]. Relative scattering intensity was converted to absolute scattering intensity by correcting the raw measured data for the contributions of the empty cell, background, sample thickness, and transmission. All the fractal behaviors and Porod models were fitted using SASfit software (Paul Scherrer Institute, Villigen, Switzerlan) [28,29,30].

#### 3.2.3. Complementary Method

The measurement of the BET specific surface area of four TATB samples was performed on a Quantachrome Autosorb-1 (Boynton, FL) and the specific surface area was determined by nitrogen adsorption from the isotherm in the relative pressure range of 0.05–0.35. The samples were degassed in vacuum at 50 °C for 2 h. 

The particle morphology and surface microstructure were investigated with the field emission scanning electron microscopy (FSEM, Ultra55, Carl, Zeiss, Germany) operating at 15 kV.

## 4. Results and Discussion

Figure 2 shows measured SANS scattering profiles for the four TATB samples as a function of contrast. The flat large Q backgrounds have been subtracted from the raw experimental data. In all cases, the changes of the intensity and the shape of the scattering curves with different contrast are visible. The changes in intensity with the contrast indicate that all the TATB powders are abundantly infiltrated by the methanol solvent. The changes in shape with the contrast exhibit different characteristics for the micro/nano-sized TATB specimens. For the micro-sized TATB, as shown in Figure 2a,b, the changes in shape of the scattering curves are subtle, probably because the size of the internal structure exceeds the detectable scale of the SANS method. For the nano-sized TATB (Figure 2c,d), however, it can be seen that each sample has varied curve shapes under different contrasts. The differences in the initial slope and deviations of power-law can reveal an obvious distribution of internal defects on the length scale of 1~100 nm, which will be discussed in detail in the following parts.

It is known that SANS data taken far from the contrast match point reflects scattering signals dominated by the external surface of the TATB crystallites [20]. Here, the data of shape function can be extracted far from the contrast match point by fitting the data from the Figure 2 according to Equation (1) to analyze the detailed shape information of the TATB crystallites. The results of this analysis are shown in Figure 3. It can be seen that all curves of TATB samples exhibit a power-law scattering of Q^−m^ (I(Q) ~ Q^−m^, where m is the Porod exponent) [31,32]. In general, the range of m-values 3 < m < 4 corresponds to surface fractal with a dimensionality Ds = 6–m between 2 and 3, the range of m-values 2 < m < 3 reflects the volume fractal with a dimensionality D_f_ = m, and there is no fractal structure when m = 4. For the micro-sized TATB samples (μTATB-1 and μTATB-2), the exponent m is found to be equal to 4 (Porod scattering) on the measured length scale (0.065~0.7 nm^−1^), indicating that the scattering surfaces of the micro-sized TATB crystals are smooth and non-fractal interfaces. The power-law exponents of the nano-sized TATB samples (nTATB-1 and nTATB-2) are observed between 3 and 4 in the measured Q range (surface fractal dimension Ds for nTATB-1 is 2.36, Ds for nTATB-2 is 2.55, 0.065~0.65 nm^−1^). Therefore, unlike the micro-sized TATB, the external surfaces of the nano-sized TATB crystallites are fractal in the law Q range. While in the large Q range (0.65~2 nm^−1^), the m-value is 4, which is the Porod scattering. However, the boundaries between the two regions are not clear. This phenomenon has also been reported by Anitas (2018) [33].

In addition, the shape functions are analyzed by using the Porod law in Figure 3, and the interfacial specific surface area of the TATB crystallites can be estimated by the relation [16]:(2)$$I\left(Q\right)=2\pi S\Delta \rho^{{}^{2}}Q^{-4},$$
where *S* is the interfacial specific surface area.

The results of this analysis in the large Q range are listed in Table 1. It is clear that the nano-sized TATB samples have much larger specific surface areas than the micro-sized TATB samples, for instance, the specific surface areas of μTATB-1 and nTATB-1 are 0.376 m^2^/g and 18.3 m^2^/g, respectively. BET experiments have also been conducted to validate the conclusions. As shown in the third column in Table 1, the specific surface areas of the nano-sized samples are much larger than the micro-sized ones, which is consistent with the results of SANS. For a clear comparison, the interfacial specific surface area of four TATB samples obtained from SANS and BET measurements are plotted in Figure 4. For the nano-sized TATB, however, the specific surface area determined by SANS is larger than that obtained from the BET method. Two reasons may explain this result. On the one hand, it is found that the nano-sized TATB particles are prone to agglomeration. After the methanol solvent was added to the TATB powders, the particles may be well expanded, thereby avoiding particle soft agglomeration. On the other hand, each measurement depends on certain assumptions. BET, which uses gaseous N_2_ adsorption, may obtain a large number of smaller pores (or cracks) and rough surfaces, but the closed pores cannot be detected. In the case of SANS, the specific surface area depends on the ability to accurately measure the concentration of the sample and the size of the surface structure being detected. Two methods provide different value; nevertheless, both methods yield a size of similar order and similar tendency of the specific surface area, confirming the feasibility of SANS to measure the specific surface area of the TATB powder.

Scanning electron microscopy (SEM) is used to observe the particle morphology and to verify the particle agglomeration issue. According to Figure 5, it can be observed that the nano-sized TATB has a more pronounced agglomeration than the micro-sized TATB, which is consistent with the SANS results. 

In general, the scattering information at the contrast match point should only arise from internal features of the TATB crystallites [20]. Figure 6 shows the scattering curves of the internal structure functions determined by fitting the SANS data to Equation (1). The four curves of the TATB samples all approximately show power-law correlation.

Figure 6a shows a comparison of the internal structure functions for the micro-sized TATB samples (μTATB-1 and μTATB-2), where the same general characteristics can be seen between the two curves. The curves have been shifted along the vertical axis (as indicated in the legend) for perspicuity. The surface fractal behavior (surface fractal dimension Ds for nTATB-1 is 2.15, Ds for nTATB-2 is 2.25.) can be observed across the entire Q range (0.06~0.5 nm^−1^) which indicates that scattering from the interfaces of internal void is rough and irregular on the measured length scale.

For the nano-sized TATB, the two curves, as shown in Figure 6b, exhibit a power-law correlation, which consists of an initial power-law fall-off in the low Q region and a transition region. The overall characteristics indicate that internal fractal surfaces exist in the nano-sized TATB over a wide range of length scale. According to previous research, it has conclusively been shown that these fractal characters may be formed from the crystallite growth on the void walls when solvent diffuses out of the system [20]. 

In the current case, the observed scattering curves over the measured Q regime (0.06 nm^−1^ < Q < 2.0 nm^−1^) can be well interpreted as scattering from two-level (small voids and large voids) structures. A diagrammatic drawing of the morphology of the internal voids of the TATB crystallites can be schematically drawn based on the above analysis, as shown in Figure 7.

In addition, quantitative information and the relevant length scale can be extracted by fitting the experimental data with the Beaucage model [34], which is given in the following equation: (3)$$\begin{array}{l} I_{Beaucage}(Q)≃G\exp(-\frac{Q^{2}R_{g}{}^{2}}{3})+B\exp(-\frac{Q^{2}R_{s}}{3}){\left(\frac{{\left[erf(QkR_{g}/\sqrt{6})\right]}^{3}}{Q}\right)}^{P}\\ +G_{s}\exp(-\frac{Q^{2}R_{s}{}^{2}}{3})+B_{s}{\left(\frac{{\left[erf(Qk_{s}R_{s}/\sqrt{6})\right]}^{3}}{Q}\right)}^{P_{s}} \end{array}$$
where *G* and *B* are Guinier and Porod factors, respectively, *R_g_* is the radius of gyration for the voids captured in the first term, *R_s_* is the radius of gyration for the sub-voids obtained in the third term, *p* is the exponent of the power-law. 

The experimental results can be well fitted to the Beaucage model assuming that the voids are spherical, as shown in Figure 6b (solid lines). These evaluated fitting parameters are displayed in Table 2. It should be noted that the internal voids of the nano-sized TATB samples are characterized by two distinct dimensions. For the nTATB-1, *R_s_* = 17.02 nm, thus the fitted diameter of the sub-voids (assuming spheres) is ca. 44 nm. While a radius of *R_g_* = 41.49 nm, indicates the average diameter of the large void (assuming spheres) is ca. 107 nm. Similar results are found for the nTATB-2 sample. 

Furthermore, the volume fraction of scatters (or the internal voids) can be calculated by neutron scattering invariant Φ*_I_* which is defined as [31]: (4)$$\Phi_{I}=\int_{0}^{\infty }Q^{2}\frac{d\Sigma }{d\Omega }(Q)dQ,$$

For a proper invariant calculation, a large Q range is generally required. In our case, the measured Q range is 0.05 nm^−1^~2 nm^−1^, making it impossible to determine the extension where the power-law behavior extends in the low Q range. Therefore, we only roughly estimate and compare the volume fraction of internal voids of different TATB samples. Values of neutron scattering invariant are given in Table 3. For the two-phase system consisting of TATB crystal and pores, volume fraction of scatters *φ* allows the calculations from the relation [31]:(5)$$\varphi \left(1-\varphi \right)=\frac{\Phi_{I}}{2\pi^{2}\left(△\rho^{2}\right)},$$

The volume fraction of four TATB samples are shown in Table 3. It depicts that the volume fractions of the nano-sized TATB samples are larger than that of micro-sized TATB samples. For instance, the volume fractions of μTATB-1 and nTATB-1 are 0.0541 and 0.128 respectively. The result may be due to the porous nature of nano-sized TATB. However, this conclusion cannot be made with certainty because the scattering range measured in the current study is limited. The size of some internal structures of micro-sized TATB may be out of the scope of the SANS measurement method.

## 5. Conclusions

SANS and contrast variation measurements were performed on two batches of micro-sized TATB and two batches of nano-sized TATB loose powders to characterize the external surface and internal voids, on length scale between 1 nm and 100 nm. Scattering results reveal that the external surface of micro-sized TATB crystallites are continuous smooth interfaces and the internal pores display a surface fractal structure (surface fractal dimension 2.15 < Ds < 2.25), while the external surface of nano-sized TATB crystallites exhibit a surface fractal behavior (surface fractal dimension 2.36 < Ds < 2.55) and their internal pores exhibit a two-level volume fractal structure (large voids consist of small voids) in the measured Q range. By using the Beaucage model, the relevant length scales are extracted. The fractal behavior extends to the maximum length scale in our SANS experiments, indicating that the fractal correlation length is on the order of 100 nm or higher. The surface area of the TATB crystallites was obtained through Porod analysis in the large Q range on the external surface. The porosity of four TATB samples were extracted by using neutron scattering invariant. The characterized results indicate that the particle sizes greatly influence the microstructural features of micro/nano-sized TATB. As the microstructural features of micro/nano-energetic materials have significant influence on their properties, this work may provide a new perspective for precisely characterizing the hierarchical structures of micro/nano-energetic materials. The obtained quantitative physical parameters can be used to establish the relationship between microstructures and properties in future study, such as developing microstructural-based simulation models of shock initiation and detonation behavior (Scaled Uniform Reactive Flow models).

## Figures and Tables

**Figure 1 materials-12-02606-f001:**
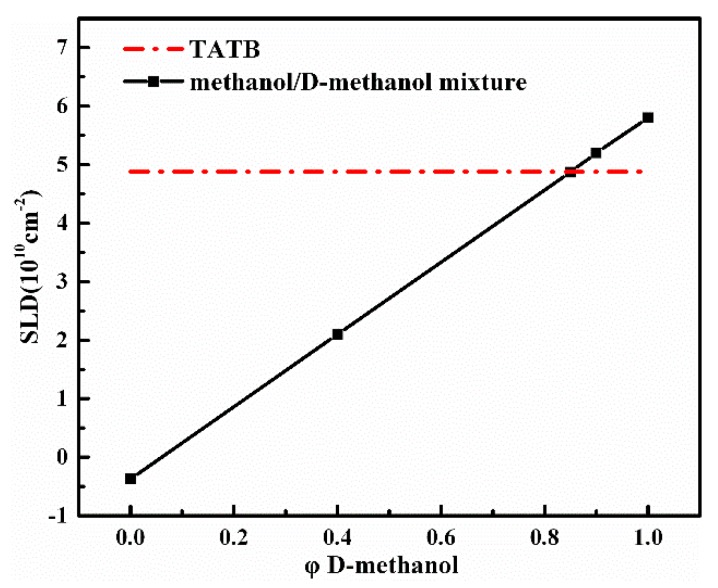
Neutron scattering length density (SLD) of the mixture of methanol and deuterated methanol (solid line) as a function of volume fraction of D-methanol, *φ*. The experimental conditions are marked by solid squares and the SLD of 1,3,5-triamino-2,4,6-trinitrobenzene (TATB) represent by dash-dotted line.

**Figure 2 materials-12-02606-f002:**
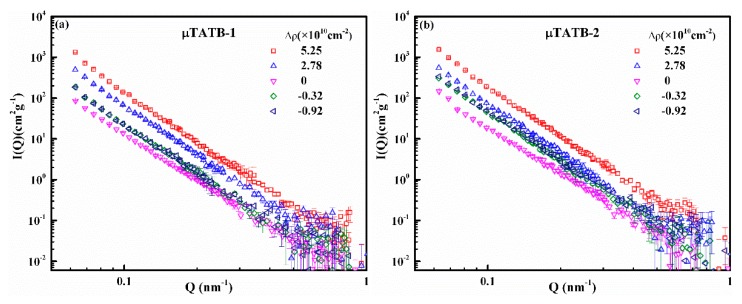
Scattering curves as a function of contrast for both micro-sized (**a**,**b**) and nano-sized (**c**,**d**) samples. Significant changes both in intensity and shape of curves are obvious for the two nano-sized samples.

**Figure 3 materials-12-02606-f003:**
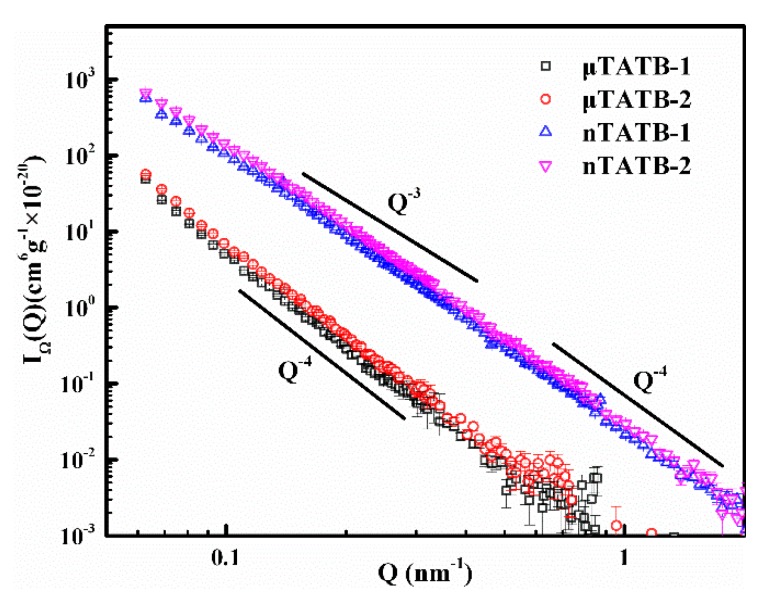
Shape function (external interfacial features) obtained far from the contrast match point. All samples show power-law scattering.

**Figure 4 materials-12-02606-f004:**
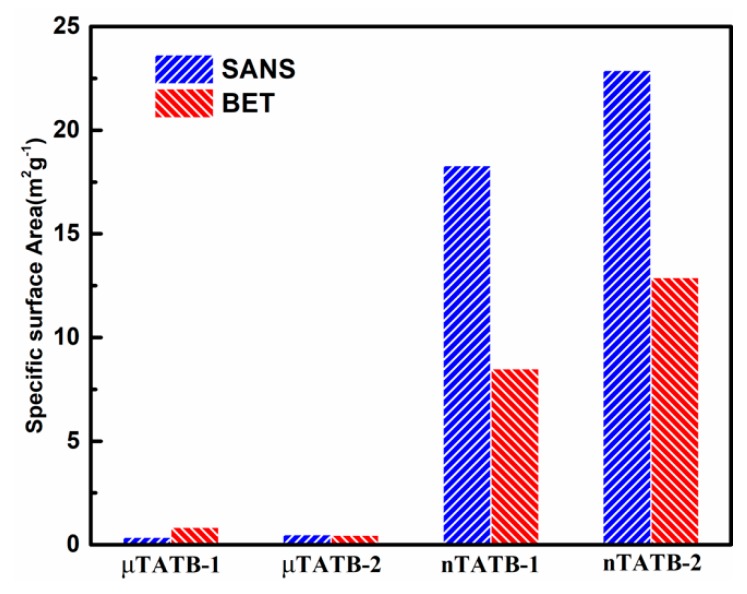
Comparison of specific surface area derived from SANS method and BET for both micro-sized and nano-sized samples (corresponding to the data listed in Table 1).

**Figure 5 materials-12-02606-f005:**
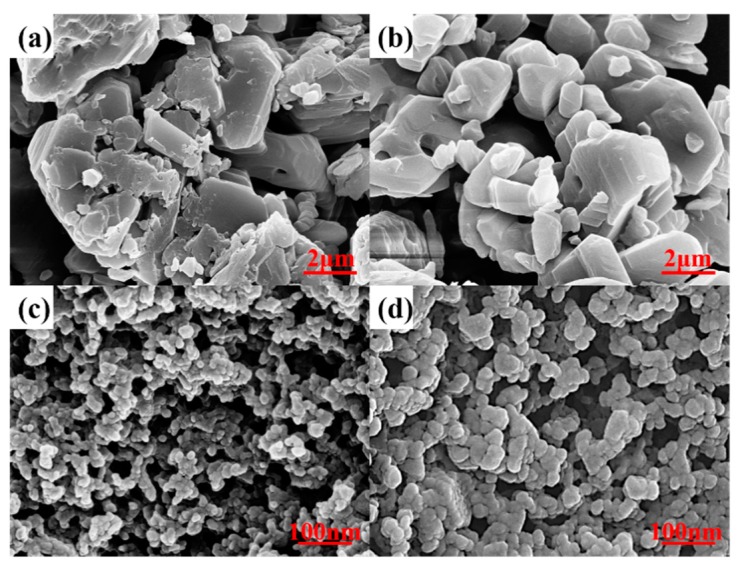
The SEM pictures of four TATB samples (**a**) μTATB-1, (**b**) μTATB-2, (**c**) nTATB-1, (**d**) nTATB-2.

**Figure 6 materials-12-02606-f006:**
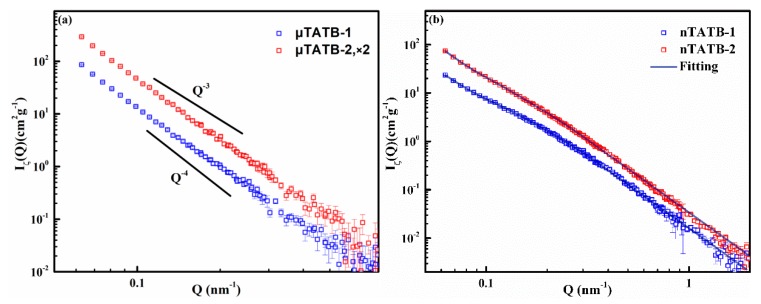
(**a**) Internal structure functions of micro-sized TATB samples. (**b**) Internal structure functions of nano-sized TATB samples, and the solid lines are the fitting results according to Equation (3).

**Figure 7 materials-12-02606-f007:**
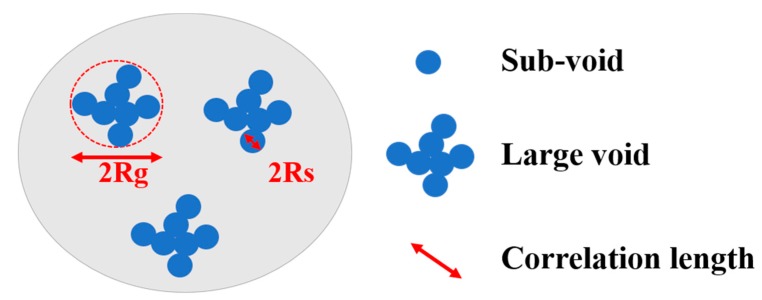
Voids composed of large voids with a radius of gyration for the entire void, *R_g_*, and a radius of gyration for the sub-voids, *R_s_*, are observed.

**Table 1 materials-12-02606-t001:** Comparison of the surface area measurements respectively by small angle neutron scattering (SANS) and Brunauer-Emmett-Teller (BET) of four representative samples.

Samples	S_SANS_ (m^2^/g)	S_BET_ (m^2^/g)	Q Range (nm^−1^)
μTATB-1	0.376	0.854	0.068~0.74
μTATB-2	0.503	0.468	0.068~0.74
nTATB-1	18.3	8.50	0.68~2.00
nTATB-2	22.9	12.9	0.68~2.00

**Table 2 materials-12-02606-t002:** Summary of fitting parameters of the nano-sized TATB at the contrast match point.

Samples	G	G_S_	B_S_	*R_g_* (nm)	*R_S_* (nm)
nTATB-1	107.708	17.5706	0.0161152	41.4934	17.0181
nTATB-2	475.460	69.0405	0.0340799	45.1318	20.2040

**Table 3 materials-12-02606-t003:** The volume fraction values using Porod Invariant of four TATB samples.

Samples	Invariant	Volume Fraction (%)
μTATB-1	0.0254	0.0541
μTATB-2	0.0434	0.0924
nTATB-1	0.0601	0.128
nTATB-2	0.169	0.361

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
