# Peer review of "Contrast Variation Small Angle Neutron Scattering Investigation of Micro- and Nano-Sized TATB"

_materials, 2019, doi:10.3390/ma12162606_

Round 1

Reviewer 1 Report

Minor comments-  

Abstract: 

Line 20  "....internal pores show the power-law correlation." is unclear. What does this correlation mean in context of small-angle scattering?

Line 23   BET is used without being defined first.

Introduction:

Line 37 "...can obviously influence the sensitivity..."  Is not obvious.  Explain. The sensitivity of what property exactly?  

Line 54  "on the other hand" doesn't make sense in this context.

Materials and Methods:

Lines 72/73   "adjusting the deuteration level of the components".   Components? or Solvent? Was TATB deuterated? or did you simply adjust the deuteration level of the solvent?

Are there exchangable hydrogens on the TATB  (the amino groups??), which will exchange with the deuterium from the Deuterated Methanol? From figure 1, it doesn't look like you accounted for this change in SLD of the TATB?  It will change what the contrast match point is.  Also, how was the concentration of the TATB estimated?

Results and Discussion:

Line 172/73   Figure 3 legend "determined the far from the contrast match point by contrast variation analysis"  doesn't make sense.  What do the authors mean? This is the basic scattering function data calculated for the solvent exclusion regions?

May wish to explain implications of power-law scattering for the scattering inexperienced audience.  What is the Porod law?

Line 197 Table 1   units are missing for the Q range column

Author Response

Response to Reviewer 1 Comments

Point 1: Line 20, internal pores show the power-law correlation." is unclear. What does this correlation mean in context of small-angle scattering?

Response 1: This part is explained in detail in the discussion section, please refer to Line 221-222. Herein, the power law correlation is about the I-Q curves from SANS, which is consisted of an initial fall-off in the low Q region and a transition region, indicating scattering from a two-level fractal structure (large voids consist of small voids). And the size of the voids is obtained by fitting the data with the Beaucage model, as shown in Eq. (3) and Fig. 6. Thanks for the Reviewer's suggestion, we have modified “their internal pores show the power-law correlation” as “their internal pores show a two-level volume fractal structure (large voids consist of small voids)”, please refer to Line 20-21.

Point 2: Line 23 BET is used without being defined first.

Response 2: BET is an acronym for three scientists: Brunauer, Emmett, and Teller. It is widely used in the powder-material field for specific surface area measurement. In most of the other articles, the BET is just used as one noun without more defining, and the BET measurement has been described in the Line 130-133, so here we use it without defining it.

Point 3: Line 37 "...can obviously influence the sensitivity..." Is not obvious. Explain. The sensitivity of what property exactly?

Response 3: Sensitivity refers to the difficulty of explosive changes when external energy is applied, which is an indicator to measure the stability of different energetic materials. The fractal characteristics can affect the sensitivity significantly. For instance, Yi et al. (reference 12) found that the fractal dimensions of nano- energetic materials are obviously lower than those of micro- energetic materials. And nano- energetic materials with low fractal dimension can significantly increase the heat conduction and heat dissipation rate, which leads to the decrease of the sensitivity.  Thanks for the Reviewer's suggestion, we have modified “Also, the fractal characteristics can obviously influence the sensitivity” as “Also, the fractal characteristics is an important factor influencing the sensitivity of different explosives”. Please refer to Line 38-39.

Point 4: Line 54 "on the other hand" doesn't make sense in this context.

Response 4: Thanks for the Reviewer's suggestion, we have modified “on the other hand” as “Also”. Please refer to Line 54.

Point 5: Lines 72/73 "adjusting the deuteration level of the components". Components? or Solvent? Was TATB deuterated? Or did you simply adjust the deuteration level of the solvent?

Response 5: Yes, we simply adjust the deuteration level of the solvent. Thanks for the Reviewer's reminding, we have changed “components” to “solvent” in the article. Please refer to Line 72.

The scattering length density of the TATB is constant, which can be calculated based on the chemical formula and the theoretical mass density.

Point 6: Are there exchangable hydrogens on the TATB (the amino groups??), which will exchange with the deuterium from the Deuterated Methanol? From figure 1, it doesn't look like you accounted for this change in SLD of the TATB? It will change what the contrast match point is. Also, how was the concentration of the TATB estimated?

Response 6: According to many literatures, TATB can only be dissolved in very concentrated acid or alkali, and is almost insoluble in common solvents (including deuterated methanol). We also known that the hydroxylic hydrogen of methanol is not so active as that of inorganic acid or alkali, so herein, we infer only a very small amount of solid-liquid interface hydroquinone isotope exchange reaction may occur under normal temperature and pressure, such as in our case of this work. Moreover, the difference of SLD between deuterated TATB and TATB (7.67×1010 cm-2 and 4.88×1010 cm-2) is not so big. Therefore, we believe the effect of hydroquinone exchange on SLD of bulk TATB powder in the current study can be ignored.

Thanks for the Reviewer's suggestion, we also plan to use some advanced techniques such as neutron diffraction or H-NMR to measure the effect of deuterated methanol soaking on hydroquinone exchange in TATB in some future work, which can provide more rigorous experimental evidence.

We did not measure the concentration of the TATB. Instead, we measured the volume fill ratio of different specimens for comparison in SANS measurement. This part was added in detail in the “Method” section.

Point 7: Line 172/73 Figure 3 legend "determined the far from the contrast match point by contrast variation analysis" doesn't make sense. What do the authors mean? This is the basic scattering function data calculated for the solvent exclusion regions?

Response 7: Due to the big difference in SLD, the scattering signal is mainly result from the external surface of the specimens when measured far from the contrast match point, just as the Reviewer's saying “It is the basic scattering function data calculated for the solvent exclusion regions”. For a clearer expression, we have modified "determined the far from the contrast match point by contrast variation analysis" to “obtained far from the contrast match point” in the article. Please refer to Line 172.

Point 8: May wish to explain implications of power-law scattering for the scattering inexperienced audience. What is the Porod law?

Response 8: According to the theory of small angle scattering, power-law describes the asymptote of the scattering intensity I(q) for large scattering wavenumbers q. The power-law, I(Q)~ Q-m (m is the Porod exponent), indicates the morphology features of a fractal system and gives the “dimensionality” of the system. For example, power-law scattering is common in self-similar or self-affine, commonly called fractal systems. In general, the range of m-values 3<m<4 corresponds to surface fractal with a dimensionality Ds = 6– m between 2 and 3, the range of m-values 2<m<3 reflects the volume fractal with a dimensionality Df = m. There is no fractal structure when m=4 and the scattering is termed Porod scattering, which is the Porod law (I(Q)~ Q-4). This part is explained in detail in the discussion section. Please refer to Line 158-162 and reference 32 .

Point 9: Line 197 Table 1 units are missing for the Q range column

Response 9: Thanks for pointing it out. It was added, please refer to Line 196.

Reviewer 2 Report

  cases of poor editing, e.g. line 84 (wrong subscript if I), line 92 (twice TATB-1), line 250 (subscript I) . Reviewer's responsibility does not include author's text editing!

SANS theory section should not be part of experimental methods

there is no description of how it was ensured that all SANS samples were of exact same TATB content. this is important as it reflects the absolute scale of scattering cross-section

I done not see significant new insight beyond reference 26.

Author Response

Response to Reviewer 2 Comments

Point 1: cases of poor editing, e.g. line 84 (wrong subscript if I), line 92 (twice TATB-1), line 250 (subscript I). Reviewer's responsibility does not include author's text editing!

Response 1: Thanks for pointing it out. All grammar and spelling issues are carefully checked and corrected.

Point 2:  SANS theory section should not be part of experimental methods

Response 2: The SANS method is a relatively rare technique for the characterization of energetic materials or for scientist about energetic materials, so a brief introduction about the SANS theory is added in order to facilitate the understanding of energetic materials for inexperienced researchers. Thanks for the Reviewer's suggestion, we have removed this section out of the experimental methods. A new section of “Theoretical background of SANS and Contrast Variation method” is added instead. Please refer to Section 2 at Line 66.

Point 3: there is no description of how it was ensured that all SANS samples were of exact same TATB content. this is important as it reflects the absolute scale of scattering cross-section

Response 3: Firstly, the volume fill ratio of all the samples are calculated before SANS. The volume fill ratio is calculated by the mass and height of the TATB added into quartz cell. Then, the scattering intensity is corrected by the fill ratio during data processing. Secondly, the area of the neutron beam is kept constant during SANS measurement.  The two treatments can ensure the effectiveness of the scattering cross section. Thanks for the Reviewer's suggestion, detailed description of the experimental procedure is added in the “Method” part. Please refer to Line 97-100.

Point 4: I done not see significant new insight beyond reference 26.

Response 4: Compared with the reference 26, we think that this article has also provided several new insights in the field of energetic materials as following.

Firstly, the object of the studied explosive material is different from that in [26].  Both micro and nano-sized TATBs are characterized by the SANS method, especially for the nano-TATB, which is the first trial in the field of nano-energetic materials to the best of our knowledge.

Secondly, the Beaucage model is used to describe structure of the inner voids for the first time, which indicates internal pores show a two-level volume fractal structure (large voids consist of small voids). Due to the colour of TATB, contrast variation SANS is the only method to characterize their inner voids, while that of RDX, HMX or CL-20 can also be observed by the optical microscopy in an index of refraction matching fluid. We believe that our analysis also privode some new perspective for precisely characterizing the hierarchical structures of micro/nano-energetic materials.

Thirdly, based on multiple analytical methods, e.g., BET specific surface area, SANS specific surface area, external surface fractal features, internal pore fractal structure, embedded porosity, etc., are extracted to characterize the micro and nano-sized TATBs, which provide a thorough understanding of the differences between the two energetic materials. We believe our results can be used as some kind of parameters to establish the relationship between microstructures and properties, such as developing the microstructure based simulation models of shock initiation and detonation behavior(Scaled Uniform Reactive Flow models).

Thanks for the Reviewer's suggestion, hope the above intentions of our manuscript can be understood.

Round 2

Reviewer 2 Report

After reading the revised manuscript and authors' comments,especially the novelty of SANS for the energetic materials community , I can recommend its publication.